# Coupling Hydrophilic Interaction Chromatography and Reverse-Phase Chromatography for Improved Direct Analysis of Grape Seed Proanthocyanidins

**DOI:** 10.3390/foods12061319

**Published:** 2023-03-20

**Authors:** Ruge Lin, Yi Wang, Huan Cheng, Shiguo Chen, Xingqian Ye, Haibo Pan

**Affiliations:** 1College of Biosystems Engineering and Food Science, National-Local Joint Engineering Laboratory of Intelligent Food Technology and Equipment, Zhejiang Key Laboratory for Agro-Food Processing, Integrated Research Base of Southern Fruit and Vegetable Preservation Technology, Zhejiang International Scientific and Technological Cooperation Base of Health Food Manufacturing and Quality Control, Zhejiang University, Hangzhou 310058, China; 2Innovation Center of Yangtze River Delta, Zhejiang University, Jiaxing 314102, China; 3Fuli Institute of Food Science, Zhejiang University, Hangzhou 310058, China; 4Ningbo Research Institute, Zhejiang University, Ningbo 315100, China; 5Zhejiang University Zhongyuan Institute, Zhengzhou 450000, China; 6Shandong (Linyi) Institute of Modern Agriculture, Zhejiang University, Linyi 276000, China

**Keywords:** proanthocyanidins, anthocyanidin, hydrophilic interaction chromatography, direct analysis, mean degree of polymerization

## Abstract

Acid-catalyzed depolymerization is recognized as the most practical method for analyzing subunit composition and the polymerization degree of proanthocyanidins, involving purification by removing free flavan-3-ols, as well as acid-catalyzed cleavage and the identification of cleavage products. However, after the removal of proanthocyanidins with low molecular weights during purification, the formation of anthocyanidins from the extension subunits accompanying acid-catalyzed cleavage occurred. Thus, grape seed extract other than purified proanthocyanidins was applied to acid-catalyzed depolymerization. Hydrophilic interaction chromatography was developed to quantify free flavan-3-ols in grape seed extract to distinguish them from flavan-3-ols from terminal subunits of proanthocyanidins. Reverse-phase chromatography was used to analyze anthocyanidins and cleavage products at 550 and 280 nm, respectively. It is found that the defects of the recognized method did not influence the results of the subunit composition, but both altered the mean degree of polymerization. The established method was able to directly analyze proanthocyanidins in grape seed extract for higher accuracy and speed than the recognized method.

## 1. Introduction

Proanthocyanidins are oligomers or polymers of monomeric flavan-3-ols joined through interflavan linkages resistant to hydrolytic cleavage, which are considered to be one of the most abundant phenolic compounds in the plant kingdom, second to lignin. They are abundant in many plant foods, such as fruits, legumes, seeds, and cereals, as well as in various drinks, such as fruit juices, wine, tea, and cider [1,2]. Previous research about proanthocyanidins was scarce because they were recognized by their ability to precipitate proteins and provide an astringent taste when being ingested from the perspective of food processing. In previous decades, proanthocyanidins have been studied extensively for their physiological activities. Epidemiological studies have provided evidence supporting the beneficial impact of dietary proanthocyanidins in scavenging free radicals and preventing relevant human diseases, such as tumors, obesity, diabetes, inflammation, and neurological disorders [3,4]. Interestingly, the close correlation between proanthocyanidins and gut microbiota has attracted the attention of scientists. Proanthocyanidins have been proven to promote the growth of beneficial bacteria and maintain intestinal microecological homeostasis [4]. Therefore, more studies have focused on the mutual effects between proanthocyanidins and gut microbiota, including how proanthocyanidins change the composition of bacteria, and which metabolites are produced from their interactions [4]. However, specific physiological activities of proanthocyanidins have not been studied thoroughly in vivo, due to the complexity of their structures [5,6].

The flavan-3-ol unit at the end of the structure of proanthocyanidins is called the terminal unit, and the other units are extension units. The structures of proanthocyanidins are determined by three factors, including linkage type, monomeric subunits, and polymerization degree. Proanthocyanidins are classified as type-A or -B according to interflavanol linkage. Type-B proanthocyanidins are linked through C4→C6 or C4→C8 bonds, while type-A are characterized by an additional C2→O7 bond. Type-B proanthocyanidins are the predominant ones in many types of plants, such as grapes, cocoa, and Chinese bayberry leaves, while type-A proanthocyanidins mainly exist in limited varieties of plants, such as cranberries, and black elderberries [7]. Different proanthocyanidins from different plants have different compositions or linkages of monomers. For instance, proanthocyanidins from grape seeds consist of (+)-catechin (C) and (−)-epicatechin (EC) mostly through type-B linkages, while proanthocyanidins from cranberries are composed of C and EC through type-A linkages [4]. Subunits, including (epi)catechin, (epi)gallocatechin, (epi)afzelechin, and their gallic acid esters, are all found in nature, and their respective polymer names are procyanidins, prodelphinidins, and propelargonidins. Among them, (epi)catechin forms the largest class of proanthocyanidins [7]. Polymerization degree commonly varies between 2 and 11 but can reach up to 50 or even more. Polymerization degree is a key feature which determines the physicochemical properties and bioavailability of proanthocyanidins [8,9]. Generally speaking, the bioavailability of proanthocyanidin dimers is only 5–10% of that of their monomers. Trimers and tetramers have lower absorption rates than dimers. Proanthocyanidins with a degree of polymerization over 4 (DP > 4) are not absorbable because of their large molecular size and gut barrier [9].

Chromatography is the most common approach to analyze the structure of proanthocyanidins. Gel permeation chromatography, size-exclusion chromatography, and normal-phase high-performance liquid chromatography are used to analyze intact proanthocyanidins, which provides information on their average molecular weight [10,11]. Hydrophilic interaction chromatography (HILIC) coupled with specific chromatography conditions has been established to separate proanthocyanidins with (epi)catechin as exclusive subunits based on their polymerization degree [12]. However, the HILIC method is not feasible to analyze proanthocyanidins containing subunits except (epi)catechin. To obtain information on linkage type, subunit composition, and polymerization degree, proanthocyanidins are purified to remove free flavan-3-ols, followed by acid-catalyzed depolymerization with nucleophiles [6,13,14]. Under acidic conditions, proanthocyanidins are depolymerized to release terminal subunits as flavan-3-ols and extension subunits as electrophilic intermediates, which can be trapped by nucleophiles to produce analyzable adducts. Flavan-3-ols and analyzable adducts reveal the linkage type and subunit composition of proanthocyanidins. Mean degree of polymerization (mDP) is calculated based on moles of terminal and extension subunits. Acid-catalyzed depolymerization with nucleophiles is advantageously applied to analyze the structure of proanthocyanidins. However, free flavan-3-ols are found to keep in purified proanthocyanidins [5,12], which increases the amount of flavan-3-ols from terminal subunits after acid-catalyzed depolymerization. In fact, electrophilic intermediates can be oxidized into anthocyanidins [15], which may competitively inhibit the nucleophilic reaction. Thus, acid-catalyzed depolymerization with nucleophiles should be improved to obtain a more accurate structure of proanthocyanidins.

Grapes are one of the richest sources of phytochemicals among commonly consumed fruits. The health benefits of the seeds or its extracts, and grape-derived products, such as wine, have been attributed to its polyphenolic compounds [16,17]. Grape seed extract is an outstanding source of polyphenols, mostly proanthocyanidins (approximately 90%) that can be found in red wine (rather than white wine) but also commercially available as capsules or tablets at different concentrations [17]. Grape seeds contain an abundance of proanthocyanidins, which are mostly dimers, trimers, and oligomers of monomeric catechins [18]. Grape seed proanthocyanidins are one of the earliest proanthocyanidins studied, because they are easy to obtain and have a mature extraction process. Therefore, grape seed proanthocyanidins will be used as an example in the present study.

Acid-catalyzed depolymerization, coupled with a HILIC analysis of free flavan-3-ols, was established to directly analyze proanthocyanidins in grape seed extract without purification. We found that proanthocyanidins were violently transformed into anthocyanidins, along with acid-catalyzed depolymerization with nucleophiles. Anthocyanidins were analyzed with reverse-phase high-performance liquid chromatography (RPHPLC) to quantify extended flavan-3-ols transformed into anthocyanidin. The comparison of the recognized method reported previously and the direct method established in the present study is shown in Figure 1.

## 2. Materials and Methods

### 2.1. Materials

Analytical standards of cyanidin chloride (≥98.0%), gallic acid (≥98.0%), (epi)catechin (≥98.0%), (epi)catechin gallate (≥98.0%), (epi)gallocatechin gallate (≥98.0%), and polystyrene were obtained from Sigma-Aldrich (St. Louis, MO, USA). Methanol, acetic acid, and tetrahydrofuran (HPLC grade) were purchased from Thermo Fisher Scientific (Fair Lawn, NJ, USA). Grape seed extract (proanthocyanidins content: 95.0%) and procyanidin B1 (≥95.0%) were purchased from Yuanye Biotechnology Co., Ltd. (Shanghai, China). Deionized water was prepared by a Millipore water purification system and used throughout the experiments. All other chemicals used were of analytical grade.

### 2.2. Purification of Proanthocyanidins from Grape Seed Extract with a Sephadex LH-20 Column

Proanthocyanidins were purified from grape seed extract using Sephadex LH-20 according to our previous report with minor modification [19]. The column (300 × 15 mm) was equilibrated with a methanol/water solution (1:1, *v*/*v*) containing 0.1% *v*/*v* trifluoroacetic acid. Grape seed extract (2.0 g) was dissolved in the mobile phase and loaded onto the column. The column was first eluted with 3 column volumes of the mobile phase to remove free flavan-3-ols, and the eluent of free flavan-3-ols was collected. Proanthocyanidins were then eluted with 3 column volumes of an acetone/water solution (2:1 *v*/*v*) containing 0.1% *v*/*v* trifluoroacetic acid, and the eluent of proanthocyanidins was collected. Both eluents were concentrated under reduced pressure at 40 °C to remove methanol and acetone, and then lyophilized to dry powder.

### 2.3. Depolymerization of Proanthocyanidins in the Presence of Excess Phloroglucinol

The depolymerization of proanthocyanidins was carried out as described previously [6]. Briefly, a solution of 0.1 M HCl in methanol, containing 50 mg/mL phloroglucinol and 10 mg/mL ascorbic acid, was prepared. Purified proanthocyanidins, or grape seed extract (5 mg), was added to 1 mL of the solution and depolymerized at 50 °C for 20 min. The reaction solution was immediately applied to an optical absorption analysis as described below. Proanthocyanidins in anhydrous methanol were set as a negative control, while proanthocyanidins in acidified methanol (0.1 M HCl) were set as a positive control. For HPLC analysis, the reaction was stopped with 5 volumes of 40 mM aqueous sodium acetate and stored immediately at −20 °C for further analysis.

### 2.4. Optical Absorption Analysis

Optical absorption of the reaction solution with a 10-fold dilution was measured over a visible band range (400 to 800 nm) using a UV-visible spectrophotometer (UV-2550, Shimadzu, Japan). Cyanidin in acidified methanol (0.1 M HCl) was also applied to optical absorption analysis. The length of the light path was 1 cm.

### 2.5. Analysis of Cleavage Products by RPHPLC-MS

Cleavage products of proanthocyanidins after depolymerization were analyzed by HPLC using a Waters e2695 HPLC system with a Waters 2489 UV-vis detector. The instrument was fitted with an Eclipse XDB-C18 column (250 mm × 4.6 mm, 5.0 μm; Agilent, Santa Clara, CA, USA) protected by an XDB-C18 guard column. Mobile phases consisted of H_2_O containing 1% acetic acid (Solvent A) and methanol containing 1% acetic acid (Solvent B). The linear gradient program was as follows: 0–10 min, 5% B; 10–30 min, 5–20% B; 30–55 min, 20–40% B. Detector was set at 280 and 550 nm. Column temperature was set up to 30 °C. Flow rate was 1.0 mL/min. Samples were filtered with a 0.22 µm membrane. Then, samples (10 µL) were injected into the column.

MS analyses were performed on a Shimadzu 8060NX mass spectrometer. The mass spectrometer was operated at a positive ion mode with an atmospheric pressure chemical ionization probe at a temperature of 450 °C and a voltage of 4.5 kV. The mass scan range was 200–1000 amu.

### 2.6. Analysis of Free Flavan-3-Ols in Grape Seed Extract by HILIC

Grape seed extract was analyzed by HILIC using a Luna HILIC column (250 × 4.6 mm, 5 μm; Phenomenex, Torrance, CA, USA). Mobile phases consisted of acetonitrile containing 0.5% acetic acid (A) and water containing 0.5% acetic acid (B). The linear gradient program was as follows: 0–30 min, 3–13% B. The detector was set at 280 for quantitative purposes, the column temperature was set up to 30 °C, and the flow rate was 0.35 mL/min. The samples were filtered with a 0.22 µm syringe filter, and then the samples (10 µL) were injected into the column.

### 2.7. Analysis of Free Flavan-3-Ols and Proanthocyanidins Removed from Grape Seed Extract by HILIC

Free flavan-3-ols and proanthocyanidins removed from grape seed extract were analyzed by HILIC using a Luna Hilic column (250 × 4.6 mm, 5 μm; Phenomenex). Mobile phases consisted of acetonitrile containing 0.5% acetic acid (A) and water containing 0.5% acetic acid (B). The linear gradient program was as follows: 0–5 min, 10–13% B; 5–20 min, 13–80% B. The detector was set at 280 nm, the column temperature was set up to 30 °C, and the flow rate was 1.0 mL/min. The samples were filtered with a 0.22 µm syringe filter, and then the samples (10 µL) were injected into the column.

### 2.8. Analysis of Molecular Weight Distribution by Gel Permeation Chromatography (GPC)

Purified proanthocyanidins after acid-catalyzed cleavage in the absence of phloroglucinol and ascorbic acid were dried under reduced pressure at 40 °C to remove methanol and HCl. The sample was applied to an analysis of molecular weight distribution by GPC according to the reported method. Purified proanthocyanidins were set as a control. The samples (3.0 mg) were suspended in acetyl bromide/glacial acetic acid (1:9, *v*/*v*) for 2 h. The solvent was then removed under reduced pressure. The residues were dissolved in tetrahydrofuran and filtered over a 0.22 μm syringe filter before injection. GPC analyses were performed with reported method [20] using a Shimadzu system, consisting of a pumping subunit (LC 20AT), a column oven (CTO-20AC), a diode array detector (SPD-M20A), a degasser subunit (DGU-20A3), and a controller subunit (CBM-20A). The Shimadzu LabSolution software package was used for system control. Three GPC columns (7.5 mm × 30 mm, 5 μm, 10,000, 1000 and 500 Å; Agilent PLgel, Santa Clara, CA, USA) were connected in a series for the analyses. Polystyrene standards with molecular weight ranging from 162 × 10^6^ to 5 × 10^6^ g/mol were used to establish calibration.

### 2.9. Determination of Proanthocyanidins with Butanol-HCl Assay

The contents of proanthocyanidins in grape seed extract and polyphenols removed by the Sephadex LH-20 column were determined with butanol-HCl assay, as reported previously [21]. Butanol-HCl reagents were prepared by dissolving 40 mg of ammonium iron (III) sulfate dodecahydrate in 3.0 mL of water, followed by adding 5.0 mL of 12 M HCl and 92 mL of n-butanol. Grape seed extract or removed polyphenols were mixed with 15 mL of butanol-HCl reagent in 25 mL glass tubes sealed with Teflon-lined screw caps. Aliquots (2 mL) of a nonheated mixture were removed to be read as controls. A reaction was performed by heating to 70 °C for 2.5 h. Their amounts were determined with a UV-visible spectrophotometer (UV-2550, Shimadzu, Japan) at 550 nm, based on the calibration established with a set of designated concentrations of procyanidin B1.

### 2.10. Statistical Analysis

Statistical analysis was performed using GraphPad Prism 9.0.0 (GraphPad Software, Boston, MA, USA). Samples were prepared and analyzed in triplicate. Data are shown as mean ± standard deviation (SD). Multiple comparisons were performed by one-way analysis of variance (ANOVA) followed by Bonferroni’s post hoc test. *p* values < 0.05 were considered statistically significant.

## 3. Results and Discussion

### 3.1. Transformation of Proanthocyanidins to Anthocyanidins along with Acid-Catalyzed Depolymerization

It is known that proanthocyanidins can be hydrolyzed into anthocyanidins in acidified 1-butanol [22]. Temperature, metal ions, and pH have a great influence on the side reaction. The cleavage of interflavan bonds in proanthocyanidins is mediated by H^+^ ions to form electrophilic intermediates from extension subunits. The electrophilic intermediates undergo structural rearrangement to form anthocyanidins. The formation of anthocyanidins may occur during acid-catalyzed depolymerization of proanthocyanidins in the presence of excess nucleophiles, such as phloroglucinol. As shown in Figure 2A, methanolic solutions of purified proanthocyanidins became red under acidic conditions, whether phloroglucinol was present or not. Both optical absorption spectra of proanthocyanidins in acidified methanol exhibited a new maximum at around 550 nm, which was consistent with the maximum of anthocyanidins. To verify the red product, an HPLC-MS analysis was performed at 550 nm. As shown in Figure 2B, an obvious peak was detected after acid-catalyzed degradation in the presence of excess phloroglucinol at 48.29 min, which was identified as cyanidin, according to the same retention time and *m/z* 287.3 [M]^+^ with cyanidin standard [23,24]. The concentration of cyanidin from extended flavan-3-ols of proanthocyanidins was 48.4 ± 3.2 μg/mL. Thus, the formation of cyanidin occurred along with acid-catalyzed depolymerization of proanthocyanidins in the presence of phloroglucinol, which would change the results of the structure analysis related to the extension subunits. The transformation of proanthocyanidins to anthocyanidins is an autoxidation mediated by metal-ion impurities in samples containing proanthocyanidins [22,25]. Thus, the yield of anthocyanidin is critically dependent on the amount of metal ion impurities. To remedy the negative impact of anthocyanidin on the structure analysis of proanthocyanidins, anthocyanidins should be analyzed to obtain complete information on the extension subunits.

### 3.2. The Effect of Cyanidin Formation on the Molecular Weight Distribution of Proanthocyanidins

Although only a small part of the proanthocyanidins was transformed into cyanidins, it may influence the structure results of proanthocyanidins, especially mDP, analyzed by acid-catalyzed depolymerization in the presence of excess phloroglucinol. To verify the assumption, GPC was used to analyze the molecular weight distributions of the starting proanthocyanidins and those after acid-catalyzed cleavage in the absence of phloroglucinol and ascorbic acid. Cyanidin and catechin standards were used to locate anthocyanidins and terminal subunits. As shown in Figure 3, it is clear that proanthocyanidins after acid-catalyzed cleavage showed a significantly lower molecular weight distribution than the starting proanthocyanidins. Apparently, this finding points towards the occurrence of depolymerization of proanthocyanidins into cyanidin and terminal flavan-3-ols with the same molecular weight as catechin. The change in the molecular weight distribution suggested that the cyanidin formation would reduce the results of mDP; however, whether it would influence the results of the subunit composition or not was unclear. Therefore, the effect of cyanidin formation on the results of subunit composition needs to be studied further.

### 3.3. Purification of Proanthocyanidins from Grape Seed Extract

Before acid-catalyzed depolymerization in the presence of excess phloroglucinol, the purification of proanthocyanidins from different materials was carried out, involving the extraction of polyphenols and the removal of free flavan-3-ols with low molecular weight [6,26,27]. However, parts of the free flavan-3-ols were not removed [12]. The residual flavan-3-ols were considered to be the flavan-3-ols from terminal subunits of proanthocyanidins, which would influence the results of the structure analysis related to terminal flavan-3-ols. Whether the proanthocyanidins are removed along with free flavan-3-ols or not has not been investigated. Here, polyphenols removed from grape seed extract by a Sephadex LH-20 column were analyzed with HILIC. Since proanthocyanidins exhibited extremely low solubility in acetonitrile, 10% of water was added to acetonitrile for full solubilization of the samples. Thus, the elution program started from 10% of water and 90% of acetonitrile, which would lead to low resolutions between free flavan-3-ols. Procyanidin B1 was used to mark the initial elution time of proanthocyanidins because it is the proanthocyanidin with the lowest molecular weight from grape seed extract. As shown in Figure 4, the free flavan-3-ols and proanthocyanidins were removed from the grape seed extract during the purification of proanthocyanidins. As determined with butanol-HCl assay, 6.63% of proanthocyanidins were removed after their purification. The peak of proanthocyanidins in removed polyphenols appeared earlier than the starting proanthocyanidins, which indicated that proanthocyanidins with low molecular weights were apt to be removed, along with free flavan-3-ols. Thus, the removal of proanthocyanidins led to the incomplete structure information of proanthocyanidins, which was analyzed by acid-catalyzed depolymerization. The influence of both the removal of proanthocyanidins and the residue of free flavan-3-ols on the structure of proanthocyanidins needs to be considered.

### 3.4. Analysis of Flavan-3-Ol Monomers in Grape Seed Extract

As the purification of proanthocyanidins would induce the removal of proanthocyanidins from grape seed extract and the residue of free flavan-3-ols, purification should not be carried out to obtain an accurate structure of proanthocyanidins analyzed with acid-catalyzed depolymerization. Therefore, free flavan-3-ols in grape seed extract should be analyzed to distinguish them from flavan-3-ols from terminal subunits of proanthocyanidins. It is hard to separate free flavan-3-ols with other polyphenols, including proanthocyanidins in grape seed extract, with RPHPLC [5,28], which makes the quantification of free flavan-3-ols inaccurate. Here, HILIC was first used to analyze free flavan-3-ols in grape seed extract. To separate free flavan-3-ols with high resolutions, the elution program must start from a low content of water. In the present study, 3% of water and 97% of acetonitrile was used as an initial elution. Since gallic acid may also be present in grape seed extract, it was analyzed along with the potential free flavan-3-ols. As shown in Figure 5, it is practical to separate free flavan-3-ols and gallic acid with resolutions of more than 1.5 (except flavan-3-ol epimers) using HILIC analysis. The peaks of flavan-3-ol epimers overlapped with each other due to the same hydrophilicity. As the molar absorption of flavan-3-ol epimers is exactly the same [6], the amounts of free flavan-3-ols required for structure analysis are calculable. The grape seed extract contained gallic acid, (epi)catechin, (epi)catechin gallate, (epi)gallocatechin gallate, and proanthocyanidins. The contents of gallic acid, (epi)catechin, (epi)catechin gallate, and (epi)gallocatechin gallate in the grape seed extract were 0.32, 42.75, 0.47, and 0.29 mg/g, respectively.

### 3.5. Direct Analysis of Proanthocyanidins in Grape Seed Extract in the Presence of Excess Phloroglucinol

To compare the recognized and direct methods for a structure analysis of proanthocyanidins, purified proanthocyanidins and grape seed extract were applied to acid-catalyzed depolymerization in the presence of excess phloroglucinol, followed by an analysis with RPHPLC. As shown in Figure 6, adduct products from purified proanthocyanidins and grape seed extract were the same, which indicated that both methods were feasible to analyze the extension subunits of proanthocyanidins. The extension subunits of proanthocyanidins consisted of catechin, epicatechin, and epicatechin gallate. The formation of anthocyanidin did not influence the result of the extension subunits. The concentrations of adduct products from purified proanthocyanidins were higher than those from grape seed extract due to their higher content of proanthocyanidins than grape seed extract. Regarding terminal subunits, the two methods obtained different concentrations of flavan-3-ols. The concentrations of catechin, epicatechin, and epicatechin gallate from purified proanthocyanidins were obviously lower than those from grape seed extract, which resulted from the removal of free flavan-3-ols through purification. The concentrations of catechin, epicatechin, and epicatechin gallate from grape seed extract were 56.9, 46.2, and 34.3 μg/mL, respectively, which were markedly higher than free (epi)catechin (36.4 μg/mL) and epicatechin gallate (4.1 μg/mL) from grape seed extract. Therefore, the terminal subunits of proanthocyanidins comprised catechin, epicatechin, and epicatechin-3-*O*-gallate, which was in agreement with the results of the recognized method.

To obtain an accurate result of mDP with the direct method, free flavan-3-ols in grape seed extract before acid-catalysis must be considered. The mDP analyzed with the direct method was calculated on a molar basis as follows:(1)$$\mathrm{mDP}=\frac{C-\mathrm{ph}+\mathrm{EC}-\mathrm{ph}+\mathrm{ECG}-\mathrm{ph}}{{\left(C+\mathrm{EC}+\mathrm{ECG}\right)}_{\mathrm{RPHPLC}}-{\left(C/\mathrm{EC}+\mathrm{ECG}\right)}_{\mathrm{HILIC}}}+1$$
where C, EC, ECG, and ph were catechin, epicatechin, epicatechin gallate, and phloroglucinol, respectively.

To remedy the negative impact of anthocyanidin formation, mDP should be calculated on a molar basis as follows:(2)$$\mathrm{mDP}=\frac{C-\mathrm{ph}+\mathrm{EC}-\mathrm{ph}+\mathrm{ECG}-\mathrm{ph}+\mathrm{cyanidin}}{{\left(C+\mathrm{EC}+\mathrm{ECG}\right)}_{\mathrm{RPHPLC}}-{\left(C/\mathrm{EC}+\mathrm{ECG}\right)}_{\mathrm{HILIC}}}+1$$

The results of the mDP obtained with the recognized and direct methods, considering anthocyanidin formation or not, were compared to analyze the effect of proanthocyanidin purification and anthocyanidin formation on mDP. As shown in Figure 7, the mDP obtained with the recognized method was significantly higher than the direct method, which indicated that purification through removal of free flavan-3-ols increased the mDP of proanthocyanidins. The formation of cyanidin exhibited different effects on the results of the mDP analyzed with the recognized and direct methods. The formation of cyanidin significantly decreased the mDP of purified proanthocyanidins, while the mDP of proanthocyanidins in grape seed extract showed a highly significant difference. This was consistent with the results of the molecular weight distribution analysis. Since the yield of cyanidin is critically dependent on the amount of metal ion impurities [22], parts of metal-ion impurities in grape seed extract were removed during the purification of proanthocyanidins, which led to the formation of less cyanidin. Even so, the formation of cyanidin might decrease the mDP of proanthocyanidins, depending on the amount of metal ion impurities in the samples.

Overall, proanthocyanidin purification and anthocyanidin formation did not influence the subunit composition of proanthocyanidins, but markedly altered their mDP. The method coupling HILIC and RPHPLC was able to directly analyze proanthocyanidins in grape seed extract for more accurate results of mDP than the recognized method.

## Figures and Tables

**Figure 1 foods-12-01319-f001:**
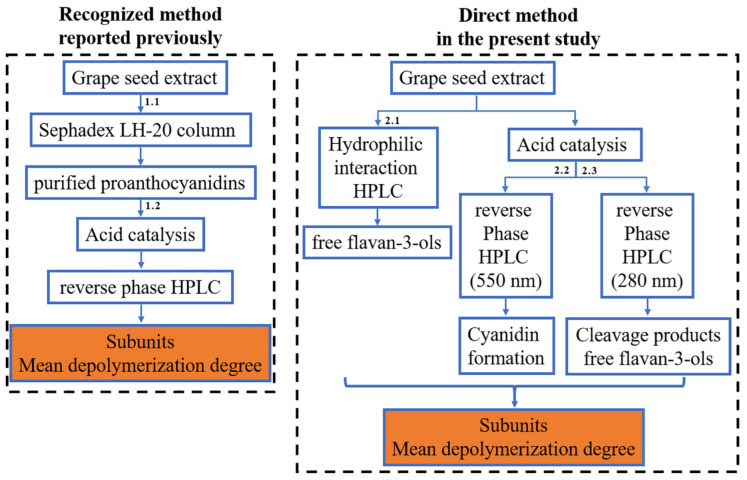
Scheme of the recognized method reported previously and the direct method established in the present study for analyzing the structure of proanthocyanidins in grape seed extract.

**Figure 2 foods-12-01319-f002:**
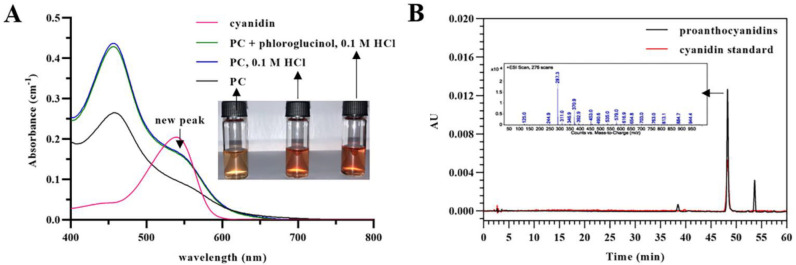
Acid-catalyzed transformation of proanthocyanidins purified from grape seed extract into cyanidin. (**A**) Optical absorption spectra of cyanidin, proanthocyanidins in pure methanol, and proanthocyanidins after acid catalysis in the presence of excess phloroglucinol or not (insets are visual images of the methanolic solutions of proanthocyanidins). (**B**) HPLC chromatograms of cyanidin standard and proanthocyanidins after acid catalysis in the presence of excess phloroglucinol at 550 nm (inset is a mass spectrum of the peak detected at 48.29 min after acid catalysis).

**Figure 3 foods-12-01319-f003:**
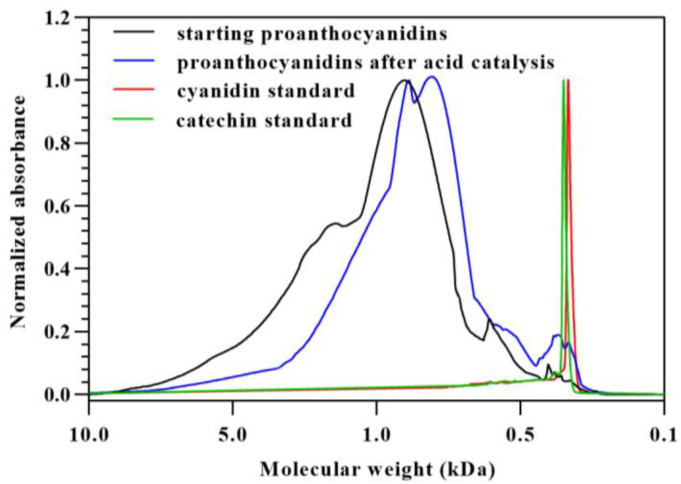
Molecular weight distributions of starting proanthocyanidins, proanthocyanidins after acid-catalyzed cleavage in the absence of phloroglucinol, cyanidin, and catechin.

**Figure 4 foods-12-01319-f004:**
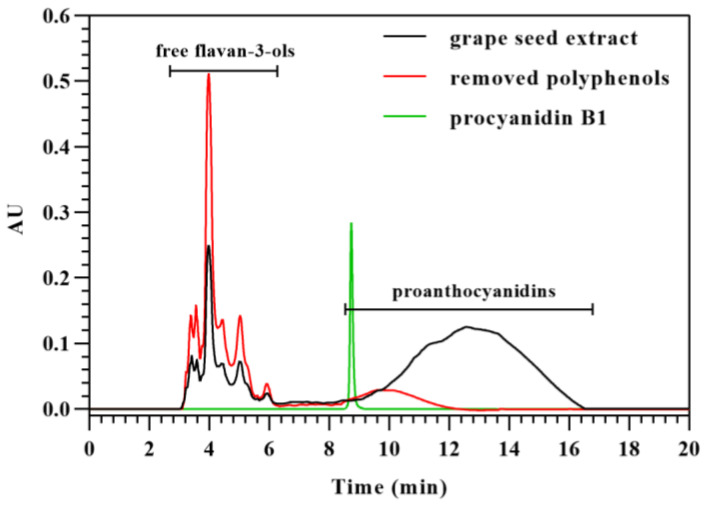
HPLC chromatograms of grape seed extract, removed polyphenols after purification of proanthocyanidins, and procyanidin B1, which were analyzed with HILIC based on their molecular weight.

**Figure 5 foods-12-01319-f005:**
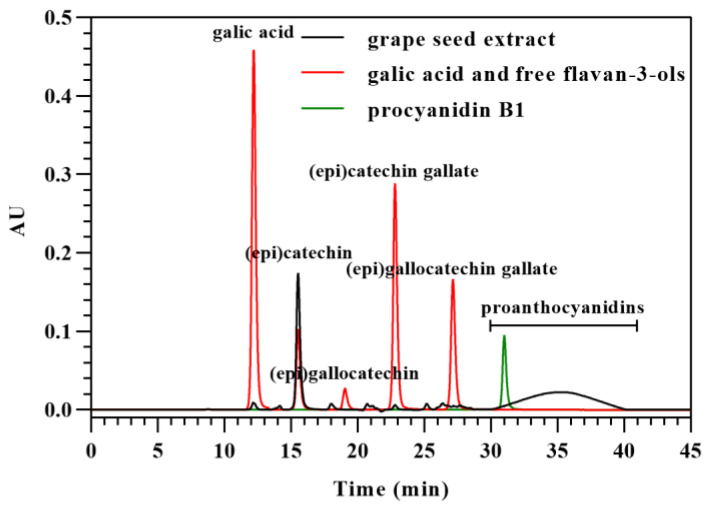
HPLC chromatograms of grape seed extract, free flavan-3-ols, and procyanidin B1, which were analyzed with HILIC based on their molecular weight.

**Figure 6 foods-12-01319-f006:**
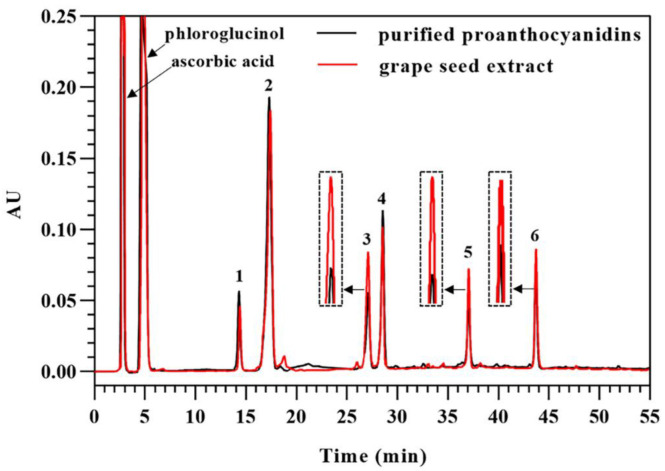
HPLC chromatograms of cleavage products from purified proanthocyanidins and grape seed extract following acid-catalysis in the presence of excess phloroglucinol. (1) catechin–phloroglucinol; (2) epicatechin–phloroglucinol; (3) catechin; (4) epicatechin gallate–phloroglucinol; (5) epicatechin; and (6) epicatechin gallate.

**Figure 7 foods-12-01319-f007:**
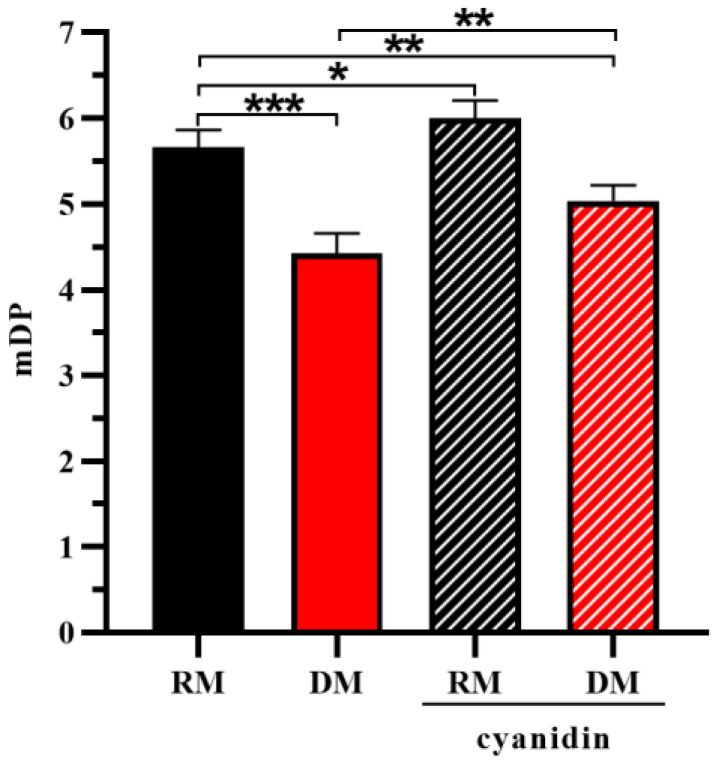
Comparison of the mDP analyzed with the recognized method (RM) and the direct method (DM), considering cyanidin formation or not. * *p* < 0.05; ** *p* < 0.01; *** *p* < 0.001.

## Data Availability

Data is contained within the article.

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
