# Peer review of "Coupling Hydrophilic Interaction Chromatography and Reverse-Phase Chromatography for Improved Direct Analysis of Grape Seed Proanthocyanidins"

_foods, 2023, doi:10.3390/foods12061319_

Round 1

Reviewer 1 Report

The paper by Lin et al. would be useful in the area that needs investigation on the structures, biological activities and effective utilization of proanthocyanidins. Only a few minor points are raised for consideration:

Page 1, lines 1 and 2: If the application of the proposed method is limited to grape seed extract, the word “grape seed” should be contained in the title.

Page 2, line 28:  The meaning of “both” is not clear to me.

Page 5, line 86: The meaning of “95.0%” is not clear. Does it mean the purity (content) of proanthocyanidins in the grape seed extract used?

Page 9, line 178: The “m/z” should be italicized.

Figure caption: The figure captions on the separate sheet are different from those in the text. Please check.

Author Response

Point 1: Page 1, lines 1 and 2: If the application of the proposed method is limited to grape seed extract, the word “grape seed” should be contained in the title.

Response 1: Thanks for the comment. “grape seed” have been added to the title.

Point 2: Page 2, line 28:  The meaning of “both” is not clear to me.

Response 2: Thanks for the comment. "Both" refers to the formation of anthocyanins and the removal of small molecular proanthocyanidins, and has been changed in the article.

Point 3: Page 5, line 86: The meaning of “95.0%” is not clear. Does it mean the purity (content) of proanthocyanidins in the grape seed extract used?

Response 3: Thanks for the comment. We confirmed with the reagent company that 95% is the content of proanthocyanidins in grape seed extract, and has been indicated in the article.

Point 4: Page 9, line 178: The “m/z” should be italicized.

Response 4: Thanks for the comment. The “m/z” has been italicized.

Point 5: Figure caption: The figure captions on the separate sheet are different from those in the text. Please check.

Response 5: Thanks for the comment. The figure captions of the whole article have been checked.

Reviewer 2 Report

In the work, the authors used reverse phase chromatography and HILIC chromatography for the determination of proanthocyanidins. The use of UV detection in my osen is not the best choice. The UV detector is not a sensitive and specific detector. Probably the authors had a difficult interpretation of the obtained spectra. A better way of detection would be to use the MS/MS detector, which is very sensitive and specific after some treatments. The authors write about the quantitative determination of analytes. Unfortunately, I do not see such results at work. This is a qualitative rather than quantitative work. The work lacks the characteristics of the methods used, e.g. LOD, LOQ, linearity range. These values would give an idea of the sensitivity of the methods used. And finally - where are the conclusions of the conducted research?

Author Response

Point 1: In the work, the authors used reverse phase chromatography and HILIC chromatography for the determination of proanthocyanidins. The use of UV detection in my osen is not the best choice. The UV detector is not a sensitive and specific detector. Probably the authors had a difficult interpretation of the obtained spectra. A better way of detection would be to use the MS/MS detector, which is very sensitive and specific after some treatments. The authors write about the quantitative determination of analytes. Unfortunately, I do not see such results at work. This is a qualitative rather than quantitative work. The work lacks the characteristics of the methods used, e.g. LOD, LOQ, linearity range. These values would give an idea of the sensitivity of the methods used. And finally - where are the conclusions of the conducted research?

Response 1: Thanks for comment. We performed a linear analysis on the HILIC analysis of the standard substances used in this paper and calculated the LLOD and LLOQ (Table 1). In addition, we measured the precision, accuracy and recovery rate of the method, and the results are shown in Table 2 and Table 3. These data prove that the UV detector we used can meet the quantitative requirements of this document, but because they are not the focus of this article, they are not included in the article. In this study, we used the coupling HILIC and RPHPLC to analyze the structure of grape seed proanthocyanidin, and confirmed that the improved method can calculate the mean degree of polymerization more accurately.

Reviewer 3 Report

Here are my suggestions (mostly technical) for authors:

1. Citations in text are not in accordance with MDPI rules. Cited references should not be in superscript but as regular numbers in squared brackets. Please revise a whole document accordingly.

2. On the page 3 in 2.2 subsection authors mentioned that, in their direct method, grape seed phenolics were purified on Sephadex but on the Figure 1 this step is not included in current but only in previously reported method? Please clarify this issue.

3. On Page 5 (subsection 2.8) there is a typo - split numerical value (7.5 mm) from unit.

4. On the same page in subsection 3.1. suggest to add word "ions" after "H+".

5. On the same page in subsection 3.1. "m/z" construction should be given in Italic style as it is usual in literature.

6. On page 6 (subsection 3.2.) I think there is a typo - you have repeated term "proanthocyanidins" two times? Check/delete surplus.

7. On page 8 (subsection 3.4.) Please specify did your results are expressed based on dry or fresh weight of seed?

8. On page 9 letter "O" in the name of epicatechin gallate should be given in Italic. Correct.

9. Please check a whole references list. Latin names for plants should be given in Italic, some titles of articles are given with capital letters for words and some not. Please check and follow MDPI instructions about this.

Kind regards.

Author Response

Point 1: Citations in text are not in accordance with MDPI rules. Cited references should not be in superscript but as regular numbers in squared brackets. Please revise a whole document accordingly.

Response 1: Thanks for comment. The citations of the whole document has been revised according to MDPI rules

Point 2: On the page 3 in 2.2 subsection authors mentioned that, in their direct method, grape seed phenolics were purified on Sephadex but on the Figure 1 this step is not included in current but only in previously reported method? Please clarify this issue.

Response 2: Thanks for comment. Sephadex LH-20 was only used in previously recognized methods, not in direct method. The material of direct method was grape seed extract without purification.

Point 3: On Page 5 (subsection 2.8) there is a typo - split numerical value (7.5 mm) from unit.

Response 3: Thanks for comment. “7.5 mm” has standardized

Point 4: On the same page in subsection 3.1. suggest to add word "ions" after "H+".

Response 4: Thanks for comment. “ions” has been after “H+”.

Point 5: On the same page in subsection 3.1. "m/z" construction should be given in Italic style as it is usual in literature.

Response 5: Thanks for comment. The “m/z” has been italicized.

Point 6: On page 6 (subsection 3.2.) I think there is a typo - you have repeated term "proanthocyanidins" two times? Check/delete surplus.

Response 6: Thanks for comment. The two “proanthocyanidins” refer to different things. The first is starting proanthocyanidins, and the second is proanthocyanidins after acid-catalyzed cleavage. To avoid ambiguity, the second “proanthocyanidins” has been changed to “those”.

Point 7: On page 8 (subsection 3.4.) Please specify did your results are expressed based on dry or fresh weight of seed?

Response 7: Thanks for comment. The result was based on dry weight of grape seed extract. Grape seed extract is dry powder, so it is not specially marked based on dry weight.

Point 8: On page 9 letter "O" in the name of epicatechin gallate should be given in Italic. Correct.

Response 8: Thanks for comment. The letter "O" in the name of epicatechin gallate has been italicized.

Point 9: Please check a whole references list. Latin names for plants should be given in Italic, some titles of articles are given with capital letters for words and some not. Please check and follow MDPI instructions about this.

Response 9: Thanks for comment. The references of the whole document have been revised according to the MDPI rules.
